# MixInYeast: A Multicenter Study on Mixed Yeast Infections

**DOI:** 10.3390/jof7010013

**Published:** 2020-12-29

**Authors:** Narda Medina, Juan Carlos Soto-Debrán, Danila Seidel, Isin Akyar, Hamid Badali, Aleksandra Barac, Stéphane Bretagne, Yasemin Cag, Carole Cassagne, Carmen Castro, Arunaloke Chakrabarti, Eric Dannaoui, Celia Cardozo, Julio Garcia-Rodriguez, Juliette Guitard, Petr Hamal, Martin Hoenigl, Tomasz Jagielski, Sadegh Khodavaisy, Giuliana Lo Cascio, María Carmen Martínez-Rubio, Joseph Meletiadis, Patricia Muñoz, Elżbieta Ochman, Teresa Peláez, Ana Perez-Ayala Balzola, Juergen Prattes, Emmanuel Roilides, Maite Ruíz-Pérez de Pipaón, Raphael Stauf, Jörg Steinmann, Ana Isabel Suárez-Barrenechea, Rocío Tejero, Laura Trovato, Lourdes Viñuela, Thanwa Wongsuk, Iwona Żak, Hossein Zarrinfar, Cornelia Lass-Flörl, Sevtap Arikan-Akdagli, Ana Alastruey-Izquierdo

**Affiliations:** 1Mycology Reference Laboratory, National Centre for Microbiology, Instituto de Salud Carlos III, 28220 Madrid, Spain; nardagab@gmail.com (N.M.); juanc.soto@isciii.es (J.C.S.-D.); 2Center for Integrated Oncology Aachen Bonn Cologne Duesseldorf (CIO ABCD), Department I of Internal Medicine, Faculty of Medicine and University Hospital of Cologne, University of Cologne, 50937 Cologne, Germany; danila.seidel@uk-koeln.de; 3European Diamond Excellence Center for Medical Mycology of the European Confederation of Medical Mycology (ECMM), 50937 Cologne, Germany; 4Cologne Excellence Cluster on Cellular Stress Responses in Aging-Associated Diseases (CECAD), University of Cologne, 50931 Cologne, Germany; 5Department of Medical Microbiology, Acibadem Mehmet Ali Aydinlar University School of Medicine, 34758 Istanbul, Turkey; isinakyar@gmail.com; 6Acibadem Labmed Laboratories, 34752 Istanbul, Turkey; 7Invasive Fungi Research Center, Communicable Diseases Institute, Mazandaran University of Medical Sciences, Sari 48157-33971, Iran; badalii@yahoo.com; 8Clinic for Infectious and Tropical Diseases, Clinical Centre of Serbia, Faculty of Medicine, University of Belgrade, 11000 Belgrade, Serbia; aleksandrabarac85@gmail.com; 9Laboratoire de Parasitologie-Mycologie, Hôpital Saint-Louis, Assistance Publique-Hôpitaux de Paris (AP-HP), 75010 Paris, France; stephane.bretagne@aphp.fr; 10Department of Infectious Agents, Université de Paris, 75006 Paris, France; 11Istanbul Medeniyet University Goztepe Training and Research Hospital, 34722 Istanbul, Turkey; yasemncag@yahoo.com; 12Department of Infectious Diseases and Clinical Microbiology, Istanbul Medeniyet University School of Medicine, 34093 Istanbul, Turkey; 13Aix-Marseille University, UMR MD3 IP-TPT, 13885 Marseilles, France; carole.cassagne@ap-hm.fr; 14Microbiology Service, Clinical Unit of Infectious Diseases and Microbiology, Hospital Universitario Valme, 41014 Sevilla, Spain; carmencmendez@hotmail.com; 15Department of Medical Microbiology, Postgraduate Institute of Medical Education and Research, Chandigarh 160012, India; arunaloke@hotmail.com; 16Unité de Parasitologie-Mycologie, Service de Microbiologie, Hôpital Européen Georges Pompidou, Assistance Publique-Hôpitaux de Paris (AP-HP) Université de Paris, 75015 Paris, France; eric.dannaoui@aphp.fr; 17Hospital Universitario Clínic, 08036 Barcelona, Spain; cgcardoz@clinic.cat; 18Hospital Universitario La Paz, 28046 Madrid, Spain; juliogarciarodriguez@gmail.com; 19Service de Parasitologie-Mycologie, Centre de Recherche Saint-Antoine, CRSA, AP-HP, Hôpital Saint-Antoine, Sorbonne Université, Inserm, 75012 Paris, France; juliette.guitard@aphp.fr; 20Department of Microbiology, Palacky University, Faculty of Medicine and Dentistry and University Hospital, 775 15 Olomouc, Czech Republic; petr.hamal@fnol.cz; 21Division of Infectious Diseases and Global Public Health, University of California San Diego, San Diego, CA 92093, USA; mhoenigl@ucsd.edu; 22Department of Medical Microbiology, Institute of Microbiology, Faculty of Biology, University of Warsaw, I. Miecznikowa 1, 02-096 Warsaw, Poland; t.jagielski@biol.uw.edu.pl; 23Division of Molecular Biology & Department of Medical Parasitology and Mycology, School of Public Health, Tehran University of Medical Sciences, Tehran 14167-53955, Iran; sadegh_7392008@yahoo.com; 24Microbiology and Virology Unit, Department of Pathology and Diagnostic, Azienda Ospedaliera Universitaria Integrata, 30126 Verona, Italy; giuliana.locascio@aovr.veneto.it; 25Hospital Universitario de Puerto Real, 11510 Cádiz, Spain; mariac.martinez.rubio.sspa@juntadeandalucia.es; 26Clinical Microbiology Laboratory, Attikon University Hospital, Medical School, National and Kapodistrian University of Athens, Haidari, 12462 Athens, Greece; jmeletiadis@med.uoa.gr; 27Clinical Microbiology and Infectious Diseases, Hospital General Universitario Gregorio Marañon, 28007 Madrid, Spain; pmunoz@hggm.es; 28Department of Medicine, Universidad Complutense de Madrid, CIBERES (CB06/06/0058), 28040 Madrid, Spain; 29Department of Clinical Microbiology, The Maria Skłodowska-Curie Institute of Oncology, W. K. Roentgena 5, 02-781 Warsaw, Poland; elzbietaochman@wp.pl; 30Hospital Universitario Central de Asturias (HUCA), Fundación para la Investigación Biomédica y la Innovación Biosanitaria del Principado de Asturias (FINBA), 33011 Asturias, Spain; mtpelaez@gmail.com; 31Microbiology Department, Hospital Universitario 12 de Octubre, 28041 Madrid, Spain; anpayala@hotmail.com; 32Section of Infectious Diseases and Tropical Medicine, Medical University of Graz, Auenbruggerplatz 15, 8036 Graz, Austria; juergen.prattes@medunigraz.at; 33Infectious Diseases Unit, 3rd Department of Pediatrics, Faculty of Medicine, Aristotle University School of Health Sciences, Hippokration General Hospital, 54642 Thessaloniki, Greece; roilides@med.auth.gr; 34Department of Infectious Diseases, Microbiology and Preventive Medicine, University Hospital Virgen del Rocío, 41013 Seville, Spain; maite.ruiz.sspa@juntadeandalucia.es; 35Institute of Clinical Hygiene, Medical Microbiology and Infectiology, Klinikum Nürnberg, Paracelsus Medical University, 90419 Nuremberg, Germany; raphael.seufert@klinikum-nuernberg.de (R.S.); Joerg.Steinmann@klinikum-nuernberg.de (J.S.); 36Institute of Medical Microbiology, University Hospital Essen, 45122 Essen, Germany; 37Infectious disease and Clinical Microbiology Department, Virgen Macarena University Hospital, 41009 Sevilla, Spain; anaisuarezb@gmail.com; 38Unit of Microbiology, Hospital Universitario Reina Sofía, 14004 Cordoba, Spain; rociotejero@hotmail.com; 39U.O.C. Laboratory Analysis Unit, A.O.U. “Policlinico-Vittorio Emanuele”, 95123 Catania, Italy; ltrovato@unict.it; 40Department of Biomedical and Biotechnological Sciences, University of Catania, 95123 Catania, Italy; 41Servicio de Microbiología Hospital Universitario Río Hortega, 47012 Valladolid, Spain; lourdesvinuela@hotmail.es; 42Department of Clinical Pathology, Faculty of Medicine Vajira Hospital, Navamindradhiraj University, Bangkok 10300, Thailand; thanwa@nmu.ac.th; 43Department of Clinical Microbiology, Children’s University Hospital of Cracow, 30-663 Kraków, Poland; bakcylka@interia.pl; 44Allergy Research Center, Mashhad University of Medical Sciences, Mashhad 91766-99199, Iran; ZarrinfarH@mums.ac.ir; 45Department of Hygiene und Medical Microbiology, Medical University of Innsbruck, 6020 Innsbruck, Austria; cornelia.lass-floerl@i-med.ac.at; 46Department of Medical Microbiology, Hacettepe University Medical School, 06100 Ankara, Turkey; sarikan@hacettepe.edu.tr

**Keywords:** yeast, chrome agar, invasive candidiasis, *Candida*, mix infections, polymicrobial infections

## Abstract

Invasive candidiasis remains one of the most prevalent systemic mycoses, and several studies have documented the presence of mixed yeast (MY) infections. Here, we describe the epidemiology, clinical, and microbiological characteristics of MY infections causing invasive candidiasis in a multicenter prospective study. Thirty-four centers from 14 countries participated. Samples were collected in each center between April to September 2018, and they were sent to a reference center to confirm identification by sequencing methods and to perform antifungal susceptibility testing, according to the European Committee on Antimicrobial Susceptibility Testing (EUCAST). A total of 6895 yeast cultures were identified and MY occurred in 150 cases (2.2%). Europe accounted for the highest number of centers, with an overall MY rate of 4.2% (118 out of 2840 yeast cultures). Of 122 MY cases, the most frequent combinations were *Candida albicans/C. glabrata* (42, 34.4%), *C. albicans/C. parapsilosis* (17, 14%), and *C. glabrata/C. tropicalis* (8, 6.5%). All Candida isolates were susceptible to amphotericin B, 6.4% were fluconazole-resistant, and two isolates (1.6%) were echinocandin-resistant. Accurate identification of the species involved in MY infections is essential to guide treatment decisions.

## 1. Introduction

Invasive candidiasis remains one of the most prevalent systemic mycoses [1,2,3]. The mortality associated with this infection is substantial, and it has been estimated to be between 10% to 47% [1,4]. *Candida albicans* is the most common species isolated, but surveillance studies have documented an increasing rate of non-*albicans* and frequently more resistant species, such as *C. glabrata* [3,5,6]. Furthermore, several studies have reported mixed yeast (MY) infections [7,8,9,10,11].

Although different microbiological media are available to detect MY cultures, some standard procedures may not be able to detect them and therefore may underestimate their burden. In one center, the detection of mixed fungemia increased from no cases to 2.8% since the introduction of chromogenic media [12]. The combination of susceptible and resistant species can complicate the clinical management, and therefore the detection of MY infections is important. We conducted a multicenter analysis to describe the epidemiology and clinical and microbiological characteristics of MY infections causing invasive candidiasis.

## 2. Materials and Methods

### 2.1. Study Design

We conducted a multicenter prospective study on invasive candidiasis caused by MY infections. The members of the ESCMID Fungal Infection Study Group (EFISG) and the Medical Mycology Study Group of the Spanish Society of Infectious Diseases and Clinical Microbiology (GEMICOMED-SEIMC) were invited to participate. In the study, 34 centers from 14 countries participated: Spain (11), France (4), Turkey (3), Iran (3), Greece (2), Italy (2), Poland (2), Germany (1), Austria (1), Czech Republic (1), India (1), Serbia (1), Thailand (1), and the United States (1). A case of MY infection was defined when 2 or more yeast species were isolated from a single culture of a normally sterile site. MY infections were detected at each participating center using a chromogenic medium. The isolates were collected prospectively between April to September 2018 and sent to the Mycology Reference Laboratory at the National Centre for Microbiology, Instituto de Salud Carlos III, Spain for further identification and susceptibility testing.

### 2.2. Data Collection

An electronic case report form (CRF) was designed using the platform clinicalsurveys.net (Questback, Cologne, Germany). All participants received the link and a personal password. Demographic, clinical, microbiological, and treatment information were recorded anonymously. For each participating center, number of primarily sterile specimens analyzed and number of sterile samples from which a yeast was isolated during the study period, were collected.

### 2.3. Molecular Identification

At the reference center (RC), isolates were cultured on 6.5% *W/V* Sabouraud Dextrose Agar (SDA) (Oxoid, Basingstoke, UK) and CHROMagar Candida medium (Oxoid, Basingstoke, UK) to visually confirm pure cultures. They were incubated at 30 °C for 24–48 h. Molecular identification was performed by polymerase chain reaction (PCR) amplifying and sequencing internal transcribed spacer regions (ITS) from the ribosomal DNA region, as previously described [13]. PCR amplicons were purified using Illustra ExoPro-Star 1-step technology (GE Healthcare Life Sciences, Buckinghamshire, UK), and subsequently sequenced by the Sanger method using a Big-Dye terminator cycle sequencing kit (Applied Biosystems, Foster City, CA, USA). DNA sequences were analyzed with DNAStar Lasergene 12 software (DNAStar Inc., Madison, WI, USA) and compared with reference sequences from the GenBank database (https://www.ncbi.nlm.nih.gov/GenBank/). Additionally, we confirmed molecular identification using the InfoQuest FP software (Bio-Rad Laboratories, Madrid, Spain) with the in-house database of the Mycology Reference Laboratory of Spain.

### 2.4. Antifungal Susceptibility

Antifungal susceptibility testing was performed according the EUCAST method [14]. The following ranges of antifungals were tested: Amphotericin B (0.03–16 mg/L) (Sigma-Aldrich, Madrid, Spain), 5-flucytosine (0.25–64 mg/L) (Sigma-Aldrich, Madrid, Spain), fluconazole (0.25–64 mg/L) (Pfizer Inc, New York, NY, USA), isavuconazole (0.015–8 mg/L) (Basilea Pharmaceutica, Basel, Switzerland), itraconazole (0.015–8 mg/L) (Janssen Pharmaceutical, Madrid, Spain), posaconazole (0.015–8 mg/L) (Merck & Co., Inc., Rahway, NJ, USA), voriconazole (0.015–8 mg/L) (Pfizer Inc, New York, NY, USA), anidulafungin (0.008–4 mg/L) (Pfizer Inc, New York, NY, USA), caspofungin (0.03–16 mg/L) (Merck &Co., Inc., Rahway, NJ, USA), and micafungin (0.004–2 mg/L) (Astellas Pharma, Inc., Tokyo, Japan). *Candida krusei* ATCC 6258 and *Candida parapsilosis* ATCC 22019 were used as quality control strains in all tests performed. The optical densities were read after 24 h. The minimal inhibitory concentrations (MIC) were defined as the lowest concentration that inhibited 90% (amphotericin B) and 50% (other antifungals) of growth. MIC values were interpreted according to EUCAST breakpoints(https://eucast.org/clinical_breakpoints/).

### 2.5. FKS Amplification and Sequencing

Hot spot regions (HS1 and HS2) of the *FKS* gene were amplified and sequenced in those isolates classified as resistant to echinocandins, as described previously [15]. DNA sequences were compared against reference sequences of *C. tropicalis* (GenBank number EU676168.2) and *C. albicans* (GenBank number XM_716336) downloaded from the GenBank database.

### 2.6. Data Analysis

Data were analyzed using SPSS Statistics 19 (SPSS Iberica, Madrid, Spain). The descriptive analysis used proportions and medians. A *p* value < 0.05 was considered significant. This study was approved by the ethical committee of the Instituto de Salud Carlos III, Madrid, Spain (Reference No. CEI PI 04_2018).

## 3. Results

### 3.1. Study Population and Clinical Characteristics

Between April and September 2018, a total of 359,686 sterile specimens were tested for yeast infections. Of these, 6895 (2%) tested positive for yeast cultures, and MY infections accounted for 150 cases (2.2%). In Europe, with 82% of the participating centers, a total of 266,579 sterile specimens were tested: 2840 (1.1%) tested positive for yeast cultures and MY infections accounted for 118 cases (4.2%). Nine centers did not report any MY cases during the study period, and one of them (from the United States) reported almost half (3157, 46%) of the total positive yeast cultures. Different rates by country were observed (Table 1). Higher rates of MY were found in Poland and France, with 6.4% and 5.6%, respectively. Austria, Serbia, Thailand, and the United States (represented by one center each) presented no cases. Among MY cases, the median age was 63 years (interquartile range: 39–73), and 53.3% were male. The patient’s information is summarized in Table 2. Frequent underlying conditions included major surgery (85, 56.6%), the use of a central vascular catheter (72, 48%), intensive care unit (ICU) stay (69, 46%), and hematological disease (53, 35.5%). A total of 95 (63.3%) cases had registered information in the CRF about antifungal administration, of which 5 (5.3%) cases received prophylaxis, 29 (30.5%) empiric treatment, and 61 targeted treatment. The overall mortality among MY cases was 29% (43 of 147 patients).

### 3.2. Species Distribution and MY Combinations

In 5 cases of the 150 MY infections, the isolates could not be recovered at the RC despite several attempts. In 10 cases, only 1 isolate was sent to the RC, and in 13 cases, the same species was identified in both isolates (Appendix A). Therefore, 249 isolates of 122 MY cases were analyzed. Five patients (4.1%) had a yeast infection caused by three species. Table 3 shows the list of species combinations. The most frequent were *C. albicans*/*C. glabrata* (42, 34.4%), *C. albicans*/*C. parapsilosis* (17, 13.9%), and *C. glabrata*/*C. tropicalis* (8, 6.6%). We found a broad diversity of combinations, and unique combinations were found in 14% of cases (17 cases out of 122). Differences in MY distribution and combinations per country are shown in Figure 1. The molecular identification detected cryptic species, such as *C. dubliniensis* or *C. orthopsilosis,* in 6.5% (8) of the cases. These cases were also identified at the participating centers by Bruker Biotyper MS matrix-assisted laser desorption ionization–time of flight mass spectrometry (MALDI-TOF MS). In 11 (9%) cases, non-*Candida* species were identified.

### 3.3. Antifungal Susceptibility Testing

Table 4 shows the geometric mean (GM), MIC ranges, MIC_50_, MIC_90_, and resistance or non-wild type (N-WT) of 245 study isolates. Four isolates did not grow after the identification. The results are shown for species with more than nine cases. According to EUCAST breakpoints, all *Candida* isolates were susceptible to amphotericin B. The MIC values of 5-flucytosine ranged from 0.120 mg/L to 64 mg/L. Fluconazole resistance was observed in 14 (6.4%) *Candida* isolates. Fluconazole-resistant *C. tropicalis* and *C. albicans* were observed in 18.2% (4 out of 22) and 4.9% (4 out of 82) of cases, respectively. As expected, *C. krusei* had elevated MICs to fluconazole (MIC_90_, 32 mg/L). The intrinsic diminished susceptibility of *C. glabrata* to fluconazole was also observed (MIC_90_, 4 mg/L), and one resistant isolate was detected (MIC_50_ 64 mg/L). The remaining fluconazole-resistant strains were *C. dubliniensis* (2), *C. inconspicua* (2), and C. *parapsilosis* (1). Azoles were less active against *C. krusei*, *C. tropicalis*, and *C. glabrata*, showing higher MIC_90_ values for itraconazole (0.5 mg/L, 0.12 mg/L, and 0.5 mg/L), and posaconazole (0.25 mg/L, 0.12 mg/L, and 0.5 mg/L), compared to *C. albicans* with 0.03 mg/L for itraconazole and posaconazole. Among the 14 fluconazole-resistant cases, treatment information was available for 7. Of those, one received fluconazole, three received echinocandins, one received voriconazole, and in two cases, the antifungal drug was not recorded.

Resistance to echinocandins was low, and two isolates were classified as resistant: One *C. albicans* and one *C. tropicalis*. Hot spot sequencing of the *FKS*1 gene showed a F641V mutation in the *C. albicans* isolate and a substitution of arginine for glycine in the seventh position of the HS1 *FKS1* in the *C. tropicalis*. MIC values for anidulafungin and micafungin were 0.06 mg/L for the *C. albicans* isolate, and 0.125 mg/L and 2 mg/L for the *C. tropicalis* isolate. The resistant *C. albicans* was isolated from a patient who had received antifungal prophylaxis with anidulafungin. The *C. tropicalis* case had not recorded prophylactic treatment. Both strains were identified in combination with a *C. parapsilosis*.

## 4. Discussion

This study described the epidemiology and clinical and microbiological characteristics of mixed yeast (MY) infections in invasive candidiasis. In this study, we found that MY occurred in 2.2% of the yeast cultures analyzed from 14 countries. The rate of MY in Europe was 4.2%. Different proportions across countries were also observed (range 0% to 6.4%). In previous studies, MY infections accounted for 1.8% to 10.6% of cases [9,10,12,16,17]. The proportion of MY found in Spain (2.4%) was slightly higher than a previous study, which detected a rate of 1.8% [12]. Other studies carried out in tertiary care hospitals in Germany and Greece have detected rates of MY of 4.4% and 4.7% [17,18], consistent with the rates detected in this study (4.9% and 5%, respectively). In contrast, in Poland, a multicenter survey in patients with candidemia detected a lower proportion of MY infections (3.5% vs. 6.4% found in this study) [16]. Similarly, the incidence of MY infections reported in a retrospective study in Turkey was 3.7% [19], while the rate in this study was found to be 4.4%. In France, rates of MY cultures of 8.7%, and 7.5% in deep-seated samples were previously described [9], slightly higher than the 6.4% obtained in this study. Furthermore, in the United States, the transplant-associated infection surveillance network reported 10.6% of mixed infections in organ transplant recipients between 2001 and 2006 [8]. However, no MY cases were detected in the participating center from the United States. The underlying conditions found in MY cases were similar to the risk factors reported for invasive candidiasis [12]. These factors included major surgery, the presence of central venous catheter, and ICU stay [3,12,20].

The combination of *C. albicans/C. glabrata* was the most frequent (34.4%). This is not surprising, since these two species are the most frequently isolated in epidemiological studies [6]. Interestingly, a murine model of oropharyngeal candidiasis (OPC) showed that colonization by *C. glabrata* was increased by co-infection or a pre-established infection with *C. albicans* [21]. The hyphal wall adhesins Als1 and Als3 of *C. albicans* were important for the in vitro adhesion of *C. glabrata* and are possible for other species [21].

Antifungal therapy is a critical component in the clinical management. However, antifungal options are limited, and drug resistance is a growing concern [6,22]. The current guidelines of the Infectious Diseases Society of America (IDSA) and the European Society of Clinical Microbiology and Infectious Diseases (ESCMID) recommend the use of an echinocandin as first-line of treatment in invasive candidiasis [4,23]. In general, the echinocandins are active drugs against most *Candida* species [24]. In this study, the overall resistance to echinocandins was low (two cases, 1.6%). In *C. glabrata*, several studies have described an increasing level of echinocandin resistance and a rapid ability of resistance development during antifungal therapy [25,26], but we did not find resistance to echinocandins in this species. Therefore, resistance should be closely monitored. In this study, the two echinocandin resistant isolates harbored *FKS1* mutations. In the *C. tropicalis* isolate, we found a substitution of arginine for a glycine in the seventh position. Different studies have also documented *C. tropicalis* isolates with echinocandin resistance and *FKS* mutations [6,27]. To our knowledge, this is the first time this mutation has been reported. In *C. albicans,* a F641V mutation was found in the *FKS*1. This mutation has already been associated with reduced susceptibility to caspofungin [28]. Interestingly, this isolate was found in a patient who received anidulafungin prophylaxis, but it was not possible to know if the patient was infected with an already resistant isolate or if the mechanism of resistance was developed during the antifungal treatment as recently investigated in other species [26].

The fluconazole-resistance rate was 6.4%, similar to the 6.9% found in a previous study in Spain [10]. In terms of patients, this rate represents 14 cases out of 122 cases (11.5%). Regarding species, the fluconazole-resistance in *C. albicans* (5.8%) was higher than those reported in monomicrobial candidemia cases in in Spain (0.9%), and Germany (0%) but was similar for *C. tropicalis* (18.2% vs. 22%) reported in Spain [10] and higher than those in Germany (12.7%) [17]. Moreover, MY combinations that included intrinsic or acquired azole-resistant strains and *C. parapsilosis* isolates might also represent a clinical concern due to the intrinsic reduced susceptibility to echinocandins of *C. parapsilosis* [29]. The overall mortality in MY infections was 29% (43 deaths of 147 cases). Other studies in patients with polymicrobial candidemia have documented mortality rates between 20% and 43% [12,20]. Although these studies did not find significant differences between cases with mixed infections and monomicrobial infections [12,20], the potential involvement of different susceptibility patterns is clinically relevant and require a rapid diagnosis.

This study has several limitations. First, as a multicentric analysis, differences in diagnostic practices might introduce variations in MY detection. Second, we estimated the proportion of MY cases over positive yeast cultures, which could lead to the underestimation of the rate of MY as one case can have several positive cultures. Differences between medical centers and the resulting patient selection may also have an influence in the results. Despite these limitations, this is the first multicentric study that has estimated the occurrence of mixed yeast infections, finding a global rate of 2.2% and 4.2% in Europe. *C. albicans/C. glabrata* was the most common combination, but a high diversity of combinations and distributions was identified. Resistance to echinocandins was present but rare and fluconazole resistance rates were variable compared with previous studies in monomicrobial infections. As different susceptibility patterns can be identified in MY infections, it is important to accurately identify the species involved and to perform susceptibility testing to support the clinical management of these infections.

## Figures and Tables

**Figure 1 jof-07-00013-f001:**
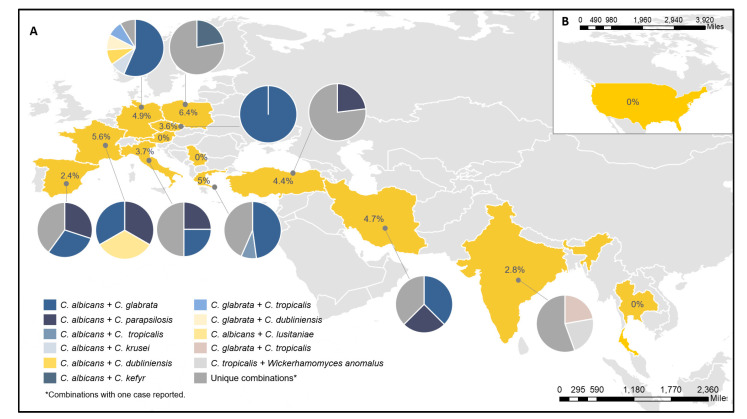
Rates, numbers, and species combinations found per country. (**A**) MY cases in Europe and Asia. (**B**) MY cases in the United States. The percentages shown inside each country represented the rate of MY per positive yeast cultures in each country and pie charts represent the MY combinations for each country.

**Table 1 jof-07-00013-t001:** Number of centers and mixed yeast (MY) proportions per country.

Country	No. Centers	No. of Yeast Cultures	Yeast Culture (+)	MY	
				*n*	% (Range)
Europe					
Poland	2	3257	188	12	6.4% (5.7%–10.0%)
France	4	75,899	444	25	5.6% (3.8%–7.5%)
Greece	2	6824	298	15	5% (0.5%–14.3%)
Germany	1	32,190	491	24	4.9% (4.9%)
Turkey	3	1853	296	13	4.4% (3.2%–7.0%)
Italy	2	17,596	164	6	3.7% (2.8%–8.7%)
Czech Republic	1	3834	56	2	3.6% (3.6%)
Spain	11	121,579	869	21	2.4% (0%–4.8%)
Austria	1	3479	23	0	0% (0%)
Serbia	1	68	11	0	0% (0%)
Total Europe	28	266,579	2840	118	4.2% (0%–6.4%)
Other countries					
Iran	3	3420	383	18	4.7% (2.8%–30%)
India	1	25,149	505	14	2.8% (2.8%)
Thailand	1	12,602	10	0	0% (0%)
The United States	1	51,936	3157	0	0% (0%)
Total	34	359,686	6895	150	2.2% (0%–6.4%)

**Table 2 jof-07-00013-t002:** Baseline characteristics and risk factors of 150 patients with MY infections.

Variable	*n*	%
Sex		
Male	80	53.3
Female	70	46.7
Age, median years (IQR)	63 (40–74)	
Underlying conditions		
Immunosuppression		
Hematological disease	53	35.5
Neutropenia	6	4.0
Solid organ transplantation	5	3.3
Major surgery	85	56.6
Chronic disease/Behavioral factor		
Alcoholism	10	6.6
Chronic cardiovascular disease	21	14.0
Chronic liver disease	10	6.6
Chronic pulmonary disease	16	10.6
Chronic renal disease	9	6.0
Diabetes mellitus	22	14.6
IV drug abuse	3	2.0
Treatment in ICU	69	46.0
Central venous catheter	72	48.0
No risk factor reported	10	6.6
Principal site of infection		
Blood	85	56.7
Peritoneal fluid	50	33.3
Biliary tract	12	8.0
Deep soft tissue	8	5.3
Lung	6	4.0
Other body sites	11	7.3

**Table 3 jof-07-00013-t003:** Species combinations of 122 MY cases.

Species Combination	*N*	%
*Candida albicans+*		
*Candida glabrata*	42	34.4%
*Candida parapsilosis*	17	13.9%
*Candida tropicalis*	5	4.1%
*Candida kefyr*	5	4.1%
*Candida krusei*	5	4.1%
*Candida dubliniensis*	3	2.5%
*Candida inconspicua*	2	1.6%
*Candida lusitaniae*	1	0.8%
*Candida orthopsilosis*	1	0.8%
*Saccharomyces cerevisiae*	1	0.8%
*Candida glabrata + Candida krusei*	1	0.8%
*Candida glabrata + Candida tropicalis*	2	1.6%
*Candida glabrata+*		
*Candida tropicalis*	8	6.6%
*Candida krusei*	3	2.5%
*Candida dubliniensis*	2	1.6%
*Candida parapsilosis*	2	1.6%
*Candida kefyr*	2	1.6%
*Candida lusitaniae*	1	0.8%
*Cyberlindnera jadinii*	1	0.8%
*Candida parapsilosis+*		
*Candida lusitaniae*	3	2.5%
*Candida tropicalis*	2	1.6%
*Lodderomyces elongisporus*	1	0.8%
*Meyerozyma guilliermondii*	1	0.8%
*Trichosporon asahii*	1	0.8%
*Candida krusei+*		
*Candida lusitaniae*	1	0.8%
*Candida tropicalis*	1	0.8%
*Candida tropicalis + Saccharomyces cerevisiae*	1	0.8%
*Dipodascus geotrichum + Pichia barkeri*	1	0.8%
*Dipodascus geotrichum*	1	0.8%
*Candida kefyr*	1	0.8%
*Candida tropicalis+*		
*Wickerhamomyces anomalus*	2	1.6%
*Candida dubliniensis*	1	0.8%
*Saccharomyces cerevisiae + Candida dubliniensis*	1	0.8%
Total	122	100%

**Table 4 jof-07-00013-t004:** In vitro antifungal activities of the isolates analyzed.

Species (No.)	AMB	5FC	FZ	ITC	ISAV	PSC	VRC	ANF	CPF	MCF
*C. albicans* (82)										
GM	0.19	0.16	0.19	0.02	0.02	0.02	0.02	0.01	0.21	0.006
MIC_50_	0.25	0.12	0.12	0.015	0.015	0.015	0.015	0.007	0.25	0.007
MIC_90_	0.25	0.25	0.25	0.03	0.06	0.03	0.015	0.015	0.25	0.007
MIC Range	0.06–0.5	0.12–64	0.12–32	0.015–1	0.015–8	0.015–1	0.015–0.25	0.007–0.060	0.004–1	0.004–0.060
R/N–WT *, *n* (%)	0 (0%)	NA	4 (4.9%)	6 (7.3%)	NA	6 (7.3%)	0 (0%)	1 (1.2%)	1 (1.2%)	1 (1.2%)
*C. glabrata* (64)										
GM	0.33	0.12	2.30	0.213	0.06	0.21	0.06	0.02	0.38	0.01
MIC_50_	0.5	0.12	2	0.25	0.06	0.25	0.06	0.015	0.5	0.007
MIC_90_	0.5	0.12	4	0.5	0.25	0.5	0.120	0.06	0.5	0.015
MIC Range	0.12–0.5	0.12–2	0.5–64	0.015–4	0.015–4	0.015–4	0.015–2	0.007–0.06	0.25–0.5	0.007–0.030
R/N–WT *, *n* (%)	0 (0%)	NA	1 (1.6%)	1 (1.6%) *	NA	2 (3.1%) *	2 (3.1%) *	0 (0%)	0 (0%)	0 (0%)
*C. parapsilosis* (27)										
GM	0.41	0.12	0.50	0.05	0.02	0.03	0.02	0.97	1.2	1.02
MIC_50_	0.5	0.12	0.50	0.06	0.015	0.03	0.015	2	1	1
MIC_90_	0.5	0.12	2	0.12	0.015	0.06	0.03	4	2	2
MIC Range	0.12–1	0.12–0.25	0.25–16	0.015–0.25	0.015–0.06	0.015–0.25	0.015–0.25	0.015–4	0.25–2	0.007–2
R/N–WT *, *n* (%)	0 (0%)	NA	1 (3.7%)	1 (3.7%)	NA	2 (7.4%)	0 (0%)	0 (0%)	0 (0%)	0 (0%)
*C. tropicalis* (22)										
GM	0.30	0.29	0.94	0.03	0.03	0.03	0.06	0.01	0.24	0.02
MIC_50_	0.25	0.12	0.5	0.03	0.015	0.03	0.5	0.015	0.25	0.015
MIC_90_	0.5	2	16	0.120	0.5	0.12	0.5	0.03	0.25	0.03
MIC Range	0.12–0.5	0.12–32	0.12–64	0.015–0.25	0.015–2	0.015–0.5	0.015–8	0.007–0.125	0.12–1	0.007–2
R/N–WT *, *n* (%)	0 (0%)	NA	4 (18.2%)	2 (9%)	NA	4 (18.2%)	3 (13.6%)	1 (4.5%)	––-	1 (4.5%) *
*C. krusei* (15)										
GM	0.48	2.64	26.60	0.14	0.08	0.25	0.25	0.03	0.66	0.09
MIC_50_	0.5	2	32	0.12	0.120	0.12	0.25	0.03	0.5	0.125
MIC_90_	0.5	4	32	0.5	0.25	0.25	0.5	0.06	1	0.125
MIC Range	0.12–0.1	1–4	16–64	0.03–0.5	0.015–0.25	0.015–0.25	0.12–0.5	0.015–0.06	0.5–1	0.007–0.12
R/N-WT *, *n* (%)	0 (0%)	NA	15 (100%) ^&^	0 (0%) *	NA	0 (0%) *	0 (0%) *	0 (0%)		0 (0%) *
Other (35)										
GM	0.26	0.30	1.05	0.08	0.04	0.07	0.04	0.05	0.40	0.06
MIC_50_	0.25	0.12	0.25	0.06	0.015	0.06	0.015	0.03	0.25	0.06
MIC_90_	0.5	4	32	0.5	0.25	0.5	0.25	4	1	1
MIC Range	0.03–1	0.12–32	0.12–64	0.015–1	0.015–1	0.015–1	0.015–1	0.007–4	0.03–16	0.007–2

AMB, amphotericin B; 5FC, 5-fluorocytosine; FZ, fluconazole; ITC, itraconazole; ISAV, isavuconazole; PSC, posaconazole; VRC, voriconazole; ANF, anidulafungin; CPF, caspofungin; MCF, micafungin. GM: geometric mean, MIC_50_: MIC that inhibits 50% of the isolates analyzed, MIC_90_: MIC that inhibits 90% of the isolates analyzed, * Based on the EUCAST epidemiological cut-off values (ECOFFs) non-wild type (N-WT); ^&^ All strains of *C. krusei* were considered fluconazole-resistant. NA: not available.

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
