# Peer review of "MixInYeast: A Multicenter Study on Mixed Yeast Infections"

_jof, 2020, doi:10.3390/jof7010013_

Round 1

Reviewer 1 Report

  1. 193, In Europe, with 82% of the participating centers. I suggest, In Europe, which accounts for an 82% of the participating centers,

This is a robust study and I only have a minor comments.

It is puzzling the absence of MY in samples form the United States. Mistakes are negated by the confirmation of the identification at the RF. Although this is commented in the Discussion, there is no cogent explanation. Shipping conditions could explain the findings? Delays due to customs clearing are not a rare event

I wonder whether hematological disease should be highlighted among the frequent underlying conditions (35,5 + 4 due to neutropenia)

Legend to Table 2. In 5 cases of the 150 MY infections the isolates could not be recovered at the RC despite several attempts. I wonder whether this refers to the inability to identify the yeasts in the reference center. If this is the case, the sentence needs clarification.

The flow chart is very useful and helps easily grasp the take-home message of the study

Author Response

Thank you for reviewing the manuscript and your suggestions for improvement.

“In Europe, with 82% of the participating centers. I suggest, In Europe, which accounts for an 82% of the participating centers”

R/ We agree with this sentence. We have changed it.

It is puzzling the absence of MY in samples from the United States. Mistakes are negated by the confirmation of the identification at the RF. Although this is commented in the Discussion, there is no cogent explanation. Shipping conditions could explain the findings? Delays due to customs clearing are not a rare event

R/ Thank you for this observation. Concerning to this frequency, we do not have any clear explanation. MY were detected at each center and then strains belonging to the MY cases were referred, in this case no MY cases were detected so strains were not referred. As mentioned in limitations differences in the type of hospitals or protocols used can have an impact but we don’t have a clear explanation for this difference.

I wonder whether hematological disease should be highlighted among the frequent underlying conditions (35,5 + 4 due to neutropenia)

R/We thank the reviewer for this comment and we agree it is an important point; we have added hematological diseases in line 202.

Legend to Table 2. In 5 cases of the 150 MY infections the isolates could not be recovered at the RC despite several attempts. I wonder whether this refers to the inability to identify the yeasts in the reference center. If this is the case, the sentence needs clarification. The flow chart is very useful and helps easily grasp the take-home message of the study

R/ Thank you for your observation. As mentioned in line 142-144 in methods, strains were sent by each center to the RC. In 5 MY cases the strains were received but they did not grow at the RC despite several attempts, so isolates could not be identified nor tested for susceptibility to antifungals.

Reviewer 2 Report

This is a nice study examining the frequency of mixed yeast (MY) infections among cases of invasive candidiasis at multiple clinical centers in multiple countries, including Europe, Middle East, Asia, and North America (United States). These data make an important contribution to our understanding of the etiology of invasive candidiasis. The methods were well described and adequately chosen, and the conclusions are generally sound. I have two suggestions.

  1. Overall these data agree reasonably well, with some allowances for inter-country and inter-hospital variability, with previous studies that examined MY infections at specific clinical sites. There is likewise a general picture emerging that a typical frequency of MY infections is 2-6% of all Candida infections. However, there is one very striking exception, which is the United States. US samples comprised nearly half of all samples studied, and yet produced zero MY cases. This is surprising and can have a number of explanations, from the different underlying candidiasis etiology to different lab practices that somehow lead to under- or over-identification of MY infections. This should be discussed in the manuscript.
  2. Table 2 shows patient characteristics for the identified MY cases, and in the discussion it is stated that "the underlying conditions found in MY cases are similar to the risk factors reported for invasive candidiasis". However, it would be useful to include a direct comparison of the MY patient population with the rest of the patient population (i.e. those infected with a single yeast) from this study, as a more relevant and informative comparator.

Minor comment: "underling" should be "underlying" throughout the manuscript.

Author Response

1. Overall these data agree reasonably well, with some allowances for inter-country and inter-hospital variability, with previous studies that examined MY infections at specific clinical sites. There is likewise a general picture emerging that a typical frequency of MY infections is 2-6% of all Candida infections. However, there is one very striking exception, which is the United States. US samples comprised nearly half of all samples studied, and yet produced zero MY cases. This is surprising and can have a number of explanations, from the different underlying candidiasis etiology to different lab practices that somehow lead to under- or over-identification of MY infections. This should be discussed in the manuscript.

 R/ Thank you for this observation. Concerning to this frequency, we do not have any clear explanation. MY were initially identified at each center and then strains were referred. As mentioned in limitations differences in type of hospitals or protocols used can have an impact but we don’t have a clear explanation for this.

2. Table 2 shows patient characteristics for the identified MY cases, and in the discussion it is stated that "the underlying conditions found in MY cases are similar to the risk factors reported for invasive candidiasis". However, it would be useful to include a direct comparison of the MY patient population with the rest of the patient population (i.e. those infected with a single yeast) from this study, as a more relevant and informative comparator.

R/ We thank the reviewer for this comment; however, we only collected clinical information of the patients with MY so comparison is not possible, taking into account the amount of infections analyzed it would have been very difficult to record clinical information of each case, so we decided to study the mixed infections and compare with previous studies that we think are already sound in candidiasis.

Minor comment: "underling" should be "underlying" throughout the manuscript.

R/ thank you for observation. We have changed it.